# Relation between Handgrip Strength and Quality of Life in Patients with Arthritis in Korea: The Korea National Health and Nutrition Examination Survey, 2015–2018

**DOI:** 10.3390/medicina58020172

**Published:** 2022-01-24

**Authors:** So-Youn Chang, Byoung-Duck Han, Kyung-Do Han, Hyo-Jin Park, Seok Kang

**Affiliations:** 1Department of Physical Medicine and Rehabilitation, Korea University Guro Hospital, Korea University College of Medicine, Seoul 08308, Korea; jsy83927@hanmail.net; 2Department of Family Medicine, Korea University Anam Hospital, Korea University College of Medicine, Seoul 02841, Korea; arybury1@naver.com; 3Department of Statistics and Actuarial Science, Soongsil University, Seoul 06978, Korea; hkd917@naver.com; 4Department of Family Medicine, Korea University Guro Hospital, Korea University College of Medicine, Seoul 08308, Korea; geeni14022@gmail.com

**Keywords:** osteoarthritis, rheumatoid arthritis, hand grip strength, quality of life (QOL)

## Abstract

*Background and Objectives*: This study aimed to investigate the relationship between handgrip strength (HGS) and quality of life (QOL) in patients diagnosed with osteoarthritis (OA) or rheumatoid arthritis (RA). *Materials and Methods*: We enrolled 13,966 from the Korea National Health and Nutrition Examination Survey from 2015 to 2018. All participants underwent the health-related QOL assessment using the European Quality of Life Scale-Five dimensions (EQ-5D) and measured the HGS. The weak HGS was defined as the lowest quartile. We investigated the difference in QOL between patients with arthritis and the healthy control group and evaluated the correlation between weak HGS and QOL in arthritis patients. *Results*: Those diagnosed with OA or RA had significantly lower QOL than healthy controls. The weak HGS was significantly correlated with lower QOL in arthritis patients. Among OA patients, those with weak HGS revealed significantly higher odds ratios for impairment in all dimensions of EQ-5D. RA patients with weak HGS had significantly higher odds ratios for impairment in dimensions of mobility, self-care, usual activity, and pain/discomfort than those with normal HGS. *Conclusions*: These results suggest that weak HGS is significantly associated with decreased QOL in patients with arthritis.

## 1. Introduction

Arthritis has a high prevalence worldwide and is a major public health problem [1,2]. Osteoarthritis (OA), also known as degenerative joint disease, is the most common type in the elderly population globally. It is characterized by gradual degeneration and loss of articular cartilage, which in turn causes secondary damage to other structures, including the synovium, meniscus, adjacent ligaments, and subchondral bone [3,4]. Rheumatoid arthritis (RA) is a systemic, chronic inflammatory autoimmune disease of unknown cause characterized by synovial inflammation, hyperplasia, autoantibody production, destruction of cartilage and bone, and systemic illnesses such as skeletal, pulmonary, cardiovascular, and psychological disorders [5]. Both diseases cause pain, stiffness, deformity, and decreased functional ability, leading to not only limitations in activities of daily living but also psychological problems [2,6].

Several studies have reported a poor quality of life (QOL) in patients with arthritis. Geryk et al. reported that the health-related quality of life (HRQOL) of patients with OA and RA was significantly worse than that of the general United States population. There were no differences between the corresponding values of patients with RA and OA [7]. In contrast, Slatkowsky-Christensen et al. reported that patients with OA and RA had weaker HRQOL than healthy controls, but there were certain differences between the two groups. Fatigue and physical function were more severe in patients with RA than in those with OA. It also showed that mental health was worse in patients with OA [8]. Lee et al. presented the effects of mental health and QOL according to sex and pain site in patients with OA [9]. Crilly et al. showed that a wide range of co-existing conditions was related to poor QOL of patients with RA [10].

Handgrip strength (HGS) measurement is a simple, inexpensive, and non-invasive overall strength assessment method for evaluating the maximal voluntary force of short duration with a dynamometer. Many studies have shown that grip strength is associated with several chronic diseases and mortality [11,12]. Sarcopenia, defined as age-related loss of skeletal muscle mass plus loss of muscle strength and/or reduced physical performance, is diagnosed using HGS, appendicular skeletal muscle mass, or the body mass index (BMI), and physical performance tests such as the gait speed and five-time chair stand test [13,14]. Moreover, HGS measurement can be useful as a concurrent value of muscle quality, functional and nutritional evaluation, general health conditions, a prognostic value of the future physical function, mortality, and hospital length of stay [11,15,16].

To our knowledge, there have been no studies investigating the relationship between muscle strength and QOL in arthritis patients. This study aimed to explore the effect of muscle strength on the QOL of patients diagnosed with arthritis. We analyzed the relationship between the QOL and HGS in arthritis patients using the Korean National Health and Nutrition Examination Survey (KNHANES) data.

## 2. Materials and Methods

### 2.1. Study Population

This cross-sectional observational study is based on the KNHANES from 2015 to 2018. As a nationally representative surveillance system, the KNHANES has been performed annually to assess the status of health and nutrition in the population of the Republic of Korea (henceforth ‘Korea’) since 1998 by the Korea Centers for Disease Control and Prevention (KCDC). The KNHANES participants were selected through multi-stage stratified clustered probability sampling. It was calculated using the sample weights assigned to the sample participants. The sample weights were constructed for sample participants representing the Korean population, taking into account the complex questionnaire design, questionnaire non-response, and post-stratification [17,18]. Participants diagnosed with OA or RA by a physician were involved in this study as patients with arthritis. As shown in Figure 1, out of 31,649 participants in the KNHANES, those aged ≥40 years were included. Patients with no data on the HGS or covariates were excluded. As a result, 13,966 individuals were enrolled in the study. We divided the participants into an arthritis group who were diagnosed with OA or RA and a healthy control group. This study was approved by the Institutional Review Board of the KCDC, and all participants provided written informed consent (No. 2018-01-03-P-A). For 2015–2017 KNHANES data, the Institutional Review Board approval was not required because the study was conducted for public welfare.

### 2.2. Demographic and Clinical Characteristics

The KNHANES was performed based on a health interview survey, health examination, and nutrition survey. The demographic and clinical characteristics of the participants analyzed in this study consisted of sociodemographic factors, anthropometric factors, comorbidities, and clinical and laboratory factors.

Sociodemographic factors included age, sex, household income, education, smoking status, drinking level, aerobic physical activity, and HRQOL. Household income was categorized into quartiles based on the national median household income and divided into low- and high-income groups. Education was divided based on a 10-year education period. Smoking status was categorized into non-smokers, former smokers, and current smokers. Non-smokers were defined as those who had smoked <5 cigarettes over their lifetime, and former smokers were defined as those who had smoked >5 cigarettes in their lifetime but not currently. The drinking level was classified into non-, low-, and high-risk drinkers. According to the Korean Ministry of Health and Welfare, high-risk drinking was defined as drinking ≥7 standard glasses for men and ≥5 standard glasses for women at least twice a week. Low-risk drinking was defined as drinking ≤4 (men) and ≤2 (women) per day. Regular exercise was defined as vigorous physical activity for ≥ 20 at a time, three times a week, or moderate physical activity for ≥30 min at a time, five times a week. The anthropometric factors included BMI (kg/m^2^), waist circumference (cm), and HGS (kg). Comorbidities included diabetes mellitus, hypertension, and hypercholesterolemia. Clinical and laboratory factors included the values for systolic blood pressure, diastolic blood pressure, and glucose, cholesterol, and high-density lipoprotein levels.

### 2.3. Measurement of HGS

HGS was measured using a digital dynamometer (T.K.K 5401; Takei Scientific Instruments Co. Ltd., Tokyo, Japan) precisely to 0.1 kg. The participant was required to stand up straight and place the forearm away from the body and rest naturally at the level of the thighs. The measurement was performed by holding the dynamometer for a maximum of 3 s without bending the elbows or wrists. The participants were instructed to hold the dynamometer as strongly as possible three times in each hand, and after each measurement, rest for at least 30 s [19]. The maximum HGS value was used for the analysis. Weak HGS was defined as the lowest quartile. The cut-off values of the HGS were 33.3 kg for men and 19.8 kg for women.

### 2.4. Evaluation of HRQOL

The European Quality of Life Scale-Five dimensions (EQ-5D) was used to evaluate health-related quality of life. The EQ-5D consists of five dimensions: mobility, self-care, usual activities, pain/discomfort, and anxiety/depression. Three levels were evaluated for each of the five dimensions: no problems, some problems, and serious problems [20]. Additionally, the EQ-5D index was used. The EQ-5D index is weighted by applying the Korean evaluation standard, and it is calculated as a weighted indicator value ranging from perfect health status (1) to the lowest score (−0.171).

### 2.5. Statistical Analysis

The demographic and clinical characteristics of the participants were presented as mean ± standard deviation for continuous variables using the Student’s *t*-test and as frequencies (percentages) for categorical variables using the chi-square test. Analysis of covariance (ANCOVA) was performed for adjustment of covariates when comparing between groups. In subgroup analysis, we analyzed patients with OA and RA separately. Bonferroni correction was performed as a post hoc test for significant results and adjusted the significance level to *p* < 0.0083. Multiple logistic regression analysis was performed to analyze the associations between the weak HGS and impaired quality of life in each dimension of EQ-5D among the groups. Model 1 was not adjusted. For model 2, age, sex, BMI, low income, education ≥ 10 years, high-risk drinking, current smoking, and aerobic physical activity were adjusted. In model 3, in addition to the modified characteristics in model 2, DM, HTN, and hypercholesterolemia levels were corrected. By comparing model 1 and model 2–3, it is possible to find out whether the adjustment of variables itself is meaningful, and in the comparison of model 2 and model 3, it can be confirmed whether there is an effect on the adjustment of comorbidities.

All statistical analyses were performed using SAS version 9.4 (SAS Institute Inc., Cary, NC, USA) considering the design of the complex sample. Statistical tests were two-sided tests with a significance level of 5%, and confidence intervals (CI) of 95% were presented for the odds ratios (OR).

## 3. Results

The basic characteristics of the arthritis and healthy control groups are presented in Table 1. The arthritis group had a significantly higher mean age and female ratio than the control group. The HGS of the arthritis group was significantly lower than the control group. The proportion of low income was significantly higher and education of more than ten years was lower in the arthritis group. Also, the rate of smokers and drinkers was significantly higher in the arthritis group. Patients with arthritis had significantly less aerobic activity than the healthy control group and had higher rates of diabetes, hypertension, and hypercholesterolemia. There were also significant differences in body mass index (BMI), waist circumference, blood pressure, and cholesterol level between the two groups. In addition, in the arthritis group, a significantly decreased QOL was observed in all five dimensions of the EQ-5D.

Table 2 shows the HGS measured in participants of the arthritis group and the control group. The HGS of arthritis patients was significantly lower than the healthy controls. However, in subgroup analysis, there was no significant difference in HGS between the OA patients and the healthy control group. The HGS of the RA patients was significantly weaker than the control group (Figure 2).

Table 3 compares the EQ-5D index according to HGS between the groups. In both arthritis and healthy control groups, the EQ-5D index was significantly lower in the participants with weak HGS. In the subgroup analysis, both OA and RA patients showed statistical significance in comparison with healthy controls using ANCOVA (*p* < 0.0001). In post hoc analysis, OA patients with weak HGS showed significantly lower QOL than normal HGS. However, there was no statistical significance in the difference of the EQ-5D index according to HGS in RA patients (Figure 3).

Multiple logistic regression analysis showed significantly increased odds ratios for impaired QOL in all dimensions of EQ-5D in arthritis patients with weak HGS than normal HGS (Table 4). Figure 4 compares the OR in subgroup analysis. The OA patients with weak HGS showed significantly higher odds ratios for impairment of all dimensions of the EQ-5D than those with normal HGS (*p* < 0.0001). In the patients with RA, there were significantly higher odds ratios for the impairment of the mobility, usual activity, self-care, and pain/discomfort dimensions in those with weak HGS than normal HGS (*p* < 0.0001).

## 4. Discussion

The aim of this study was to investigate the relationship between muscle strength and QOL in arthritis patients. The results suggest that a decrease in the QOL of arthritis patients is related to muscle strength. Compared to healthy controls, patients with arthritis had significantly lower HGS, which was more evident in patients with RA. In the analysis using EQ-5D, the QOL of arthritis patients with weak grip strength was significantly reduced compared to those with normal HGS. There was a significant association between muscle weakness and impaired QOL or arthritis patients in all dimensions of EQ-5D.

HGS is an important assessment tool for diagnosing and evaluating sarcopenia. There have been many studies on the relationships between HGS and QOL. Park et al. reported a significant relationship between weak HGS and quality of life among Korean men and women over 20 years [21]. Halaweh et al. and Kim et al. demonstrated a correlation between QOL and HGS in older adults [22,23]. In the study of Rashed et al., HGS in children and adolescents with juvenile idiopathic arthritis was confirmed as an independent predictor of disease activity, disability, and QOL and showed significant value as a simple, non-invasive tool [24]. The present study demonstrates that the HGS of Korean arthritis patients over 40 years has a significant relationship with the QOL. However, in arthritis, the handgrip strength may not only reflect muscle weakness or sarcopenia but can be affected by joint deformity, pain and swelling.

Through subgroup analysis in this study, we confirmed a difference in the relationship between QOL and HGS between OA and RA patients. There was no statistically significant difference in HGS between OA patients and healthy controls. However, OA patients with weak HGS showed significantly impaired QOL than patients with normal HGS. Multiple logistic regression analysis results revealed that OA patients with weak HGS had considerably higher odds ratios for the impairment of QOL in all five dimensions of EQ-5D than those with normal HGS. These results could suggest that muscle strength is a considerably important factor affecting the impairment of QOL in OA.

In contrast to OA, RA patients showed significantly weak HGS than the healthy controls. This finding seems to be related to that the most susceptible areas in RA are the hands and wrists, unlike OA, which commonly affects the weight-bearing joints such as the knee and hip [25,26]. The decrease in QOL according to HGS was not statistically significant in RA patients. However, compared to the healthy controls belonging to the same HGS group, the QOL of RA patients was significantly impaired. RA is a systemic chronic inflammatory autoimmune disorder, and patients with RA could have co-existing conditions such as extra-articular features [10]. The QOL of RA patients might be affected by these diverse conditions. Nonetheless, in multiple logistic regression analysis, RA patients with weak HGS showed significantly higher odds ratios than those with normal HGS in several dimensions of EQ-5D, including mobility, self-care, usual activity, and pain/discomfort. Thus, muscle strength should be considered one of the essential factors affecting the QOL of RA patients.

As mentioned earlier, OA usually affects mobility, mainly using weight-bearing joints such as the hip and knee joints, and RA commonly occurs in the hand and wrist, affecting self-care [25,26]. In the present study, the multiple logistic regression analysis has confirmed that these properties could be exacerbated due to muscle weakness. Mobility was the dimension with the highest OR (4.591, 95% CI 3.708–5.683) for deterioration of QOL in the OA patients with weak HGS. In RA patients, self-care was the dimension in which the OR was highest (4.328, 95% CI 2583–7251) in those with weak HGS.

Many studies have shown that exercise is effective in patients with arthritis. For patients with OA, exercise was strongly recommended for the hand, knee, and hip. In patients with RA increased physical activity showed improvement of the symptoms and a reduction of the impact of systemic manifestations [27,28]. Therefore, it can be concluded that exercise contributes to improving the QOL of patients with arthritis by increasing muscle strength. In addition, the weak HGS in arthritis patients is possibly associated with joint deformity or pain, not only muscle strength. Thus, management of these symptoms also could be helpful to increase the QOL of the patient by improving the HGS.

A limitation of this study is that the causality between the HGS and QOL cannot be identified in a cross-sectional design based on the National Health and Nutrition Examination Survey. First, reduced physical function, represented by weak HGS, may be the cause of impaired QOL. In contrast, arthritis patients with good QOL may be better managed and more independent; hence, their physical function is maintained well. Second, in this study, the cut-off value of the HGS was arbitrarily determined. We defined the 1st quartile (<25th percentile) as weak HGS and the 2nd-4th quartiles as normal HGS. The cut-off values of the HGS were 33.3 kg for men and 19.8 kg for women in this study. However, it is different from the cut-off value (30 kg for men and 20 kg for women) of the HGS of the sarcopenia diagnostic criteria established by the European Working Group on Sarcopenia in Older People [29]. Third, we could not obtain detailed medical information related to the arthritis site, its severity, and whether surgery was performed or not. Fourth, EQ-5D was used as a health-related QOL evaluation tool in this study. There could be inconsistencies with other studies evaluating QOL in arthritis patients using other assessment tools, including SF-36, SF-12, etc. [7,8]. Finally, since our data were based on the Korean population, it was difficult to generalize our results to other ethnic populations.

Nevertheless, the advantage of this study is that it encompassed a large sample size of 13,966 participants and that various potential confounding variables were corrected. In addition, it is meaningful that the relationship between the HGS and QOL was investigated for patients with OA and RA and explored in detail according to the five dimensions of the EQ-5D to determine which dimension was the most affected.

## 5. Conclusions

There was a significant association between weak HGS and decreased QOL in patients with arthritis. Improving muscle strength and management of joint deformity or pain associated with weak HGS may help increase the QOL of arthritis patients.

## Figures and Tables

**Figure 1 medicina-58-00172-f001:**
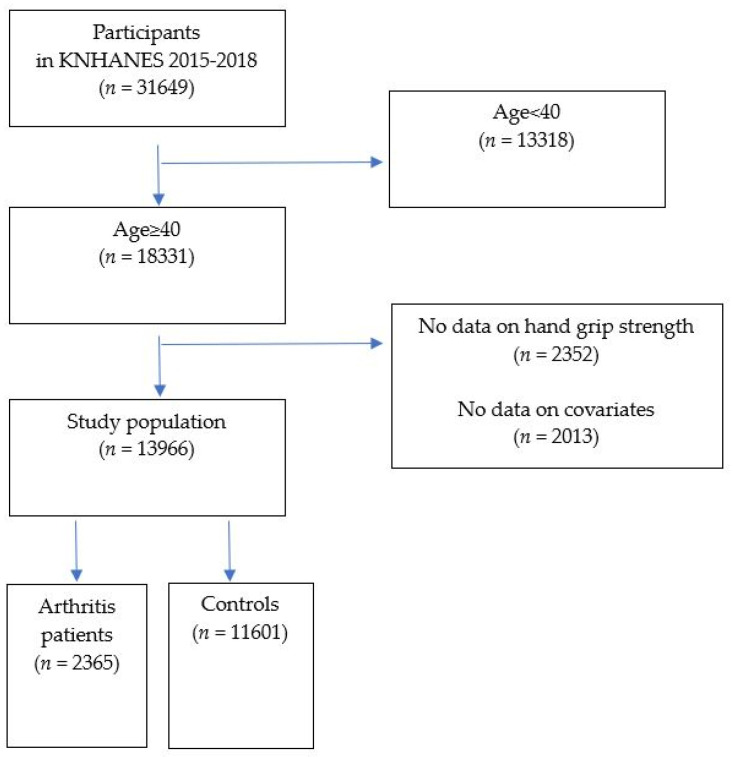
Flow diagram of participant selection.

**Figure 2 medicina-58-00172-f002:**
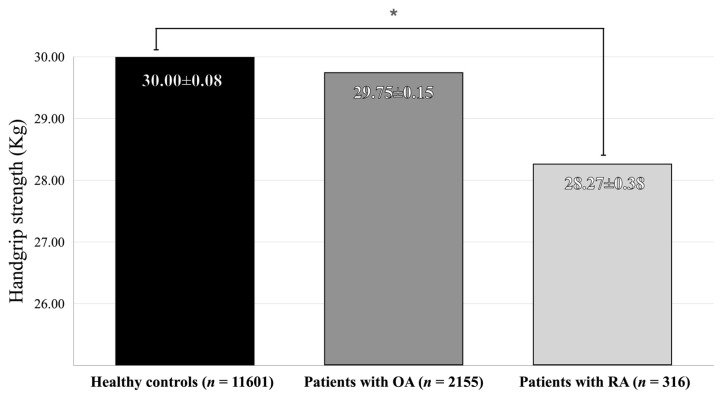
Comparison of handgrip strength between the patients with osteoarthritis or rheumatoid arthritis and healthy controls. Abbreviations: OA osteoarthritis, RA rheumatoid arthritis. Adjusted for age, sex, BMI, low income, education ≥ 10 years, high-risk drinking, current -smoking, aerobic physical activity, DM, HTN, and hypercholesterolemia levels. *p*-values were analyzed by ANCOVA. * *p* < 0.0001.

**Figure 3 medicina-58-00172-f003:**
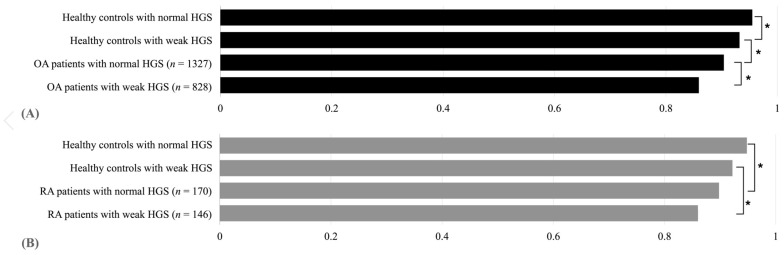
Subgroup analysis of EQ-5D index depending on HGS for patients with osteoarthritis (**A**) and patients with rheumatoid arthritis (**B**). Abbreviations: EQ-D5 European Quality of Life Scale-Five, HGS handgrip strength, OA osteoarthritis, RA rheumatoid arthritis. Adjusted for age, sex, BMI, low income, education ≥ 10 years, high-risk drinking, current smoking, aerobic physical activity, DM, HTN, and hypercholesterolemia levels. *p*-values were analyzed by Bonferroni post-test after ANCOVA. * *p* < 0.0083 between the two groups.

**Figure 4 medicina-58-00172-f004:**
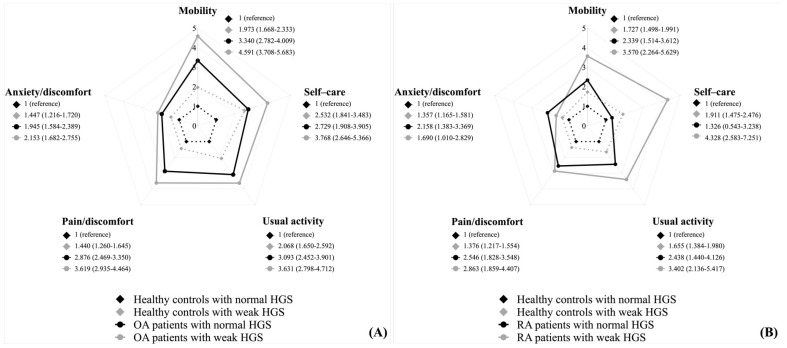
Associations between weak handgrip strength and impaired quality of life (the five dimensions of the EQ-5D) in patients with osteoarthritis (**A**) and patients with rheumatoid arthritis (**B**). Abbreviations: EQ-D5 European Quality of Life Scale-Five, HGS handgrip strength, OA osteoarthritis, RA rheumatoid arthritis, OR odds ratio, CI confidence interval. Adjusted for age, sex, BMI, low income, education ≥ 10 years, high-risk drinking, current smoking, aerobic physical activity, DM, HTN, and hypercholesterolemia levels. Values are presented as OR (95% CI). Analyzed by multiple logistic regression analysis.

**Table 1 medicina-58-00172-t001:** Demographic and clinical characteristics of participants.

Characteristics	Arthritis (*n* = 2365)	Control (*n* = 11,601)	*p*-Value
Age (years)	65.06 ± 0.25	54.87 ± 0.16	<0.0001 *
Sex			<0.0001 *
Male	21.79 (0.97)	53.83 (0.45)	
Female	78.21 (0.97)	46.17 (0.45)	
HGS (kg)	24.39 ± 0.2	32.54 ± 0.12	<0.0001 *
Weak HGS	36.19 (1.16)	18.84 (0.5)	<0.0001 *
Low income	36.93 (1.26)	14.47 (0.54)	<0.0001 *
Education ≥ 10 years	33.47 (1.25)	71.71 (0.7)	<0.0001 *
Smoking			<0.0001 *
Non smoker	77.39 (1.05)	53.68 (0.5)	
Ex-smoker	14.77 (0.88)	25.6 (0.45)	
Current smoker	7.84 (0.73)	20.72 (0.5)	
Drinking			<0.0001 *
Non	44.35 (1.23)	24.97 (0.52)	
Low risk	52.36 (1.25)	65.1 (0.57)	
High risk	3.29 (0.48)	9.93 (0.32)	
Aerobic physical activity	34.07 (1.19)	44.29 (0.63)	<0.0001 *
Diabetes mellitus	19.75 (1.04)	13.86 (0.38)	<0.0001 *
Hypertension	52.77 (1.19)	35.2 (0.58)	<0.0001 *
Hypercholesterolemia	36.12 (1.11)	24.26 (0.49)	<0.0001 *
Body mass index (kg/m^2^)	24.93 ± 0.08	24.06 ± 0.04	<0.0001 *
Waist circumference (cm)	85.25 ± 0.24	83.38 ± 0.11	<.0001 *
Systolic blood pressure (mmHg)	125.63 ± 0.4	120.15 ± 0.21	<0.0001 *
Diastolic blood pressure (mmHg)	74.99 ± 0.24	77.35 ± 0.13	<0.0001 *
Glucose	104.54 ± 0.63	103.56 ± 0.28	0.1489
Cholesterol	193.19 ± 0.97	196.39 ± 0.44	0.0022 *
High-density lipoprotein	50.03 ± 0.32	50.13 ± 0.14	0.7748
EQ-5D			
Mobility	40.53 (1.15)	9.37 (0.33)	<0.0001 *
Self-care	9.09 (0.66)	2.2 (0.15)	<0.0001 *
Usual activities	21.16 (0.95)	4.84 (0.23)	<0.0001 *
Pain/discomfort	49.84 (1.21)	18.24 (0.43)	<0.0001 *
Anxiety/depression	18.17 (0.95)	7.13 (0.29)	<0.0001 *

Abbreviations: HGS: handgrip strength. EQ-5D European Quality of Life Scale-Five. Values are presented as mean ± SD or estimated % (standard error). *p*-values were analyzed by student *t*-test or χ2 test. * *p* < 0.05.

**Table 2 medicina-58-00172-t002:** Handgrip strength measurement in patients with arthritis and healthy controls.

	*n*	Model 1 ^†^	Model 2 ^‡^	Model 3 ^§^
Controls	11,601	32.54 ± 0.12	29.99 ± 0.08	30 ± 0.08
Patients with arthritis	2365	24.39 ± 0.2	29.56 ± 0.14	29.54 ± 0.15
*p*-value		<0.0001 *	0.0049 *	0.0035 *

Values are presented as mean ± SD or number. *p*-values were analyzed by ANCOVA. * *p* < 0.05. ^†^ Not adjusted. ^‡^ Adjusted for age, sex, BMI, low income, education ≥ 10 years, high-risk drinking, current smoking, and aerobic physical activity. ^§^ Adjusted for age, sex, BMI, low income, education ≥ 10 years, high-risk drinking, current smoking, aerobic physical activity, DM, HTN, and hypercholesterolemia levels.

**Table 3 medicina-58-00172-t003:** Comparisons of EQ-5D index depending on HGS in patients with arthritis and healthy controls.

	*n*	Model 1 ^†^	Model 2 ^‡^	Model 3 ^§^
Healthy controls with normal HGS	9012	0.971 ± 0.001	0.956 ± 0.001	0.956 ± 0.001
Healthy controls with weak HGS	2589	0.925 ± 0.003	0.933 ± 0.003	0.933 ± 0.003
Arthritis patients with normal HGS	1438	0.889 ± 0.004	0.905 ± 0.004	0.905 ± 0.004
Arthritis patients with weak HGS	927	0.829 ± 0.007	0.862 ± 0.006	0.862 ± 0.006
*p*-value		<0.0001 *	<0.0001 *	<0.0001 *

Abbreviations: EQ-5D European Quality of Life Scale-Five, HGS handgrip strength. Values are presented as mean ± SD or number. *p*-values were analyzed by ANCOVA. * *p* < 0.05. ^†^ Not adjusted. ^‡^ Adjusted for age, sex, BMI, low income, education ≥ 10 years, high-risk drinking, current smoking, and aerobic physical activity. ^§^ Adjusted for age, sex, BMI, low income, education ≥ 10 years, high-risk drinking, current smoking, aerobic physical activity, DM, HTN, and hypercholesterolemia levels.

**Table 4 medicina-58-00172-t004:** Associations between weak HGS and impaired quality of life (five dimensions of the EQ-5D) in patients with arthritis and healthy controls.

Dimension of the EQ-5D	Groups	Model 1 ^†^	Model 2 ^‡^	Model 3 ^§^
Mobility	Healthy control with normal HGS	1 (ref.)	1 (ref.)	1 (ref.)
	Healthy control with weak HGS	4.202 (3.642–4.847)	1.976 (1.663–2.347)	1.969 (1.659–2.337)
	Arthritis patients with normal HGS	7.521 (6.447–8.774)	3.351 (2.811–3.994)	3.367 (2.823–4.015)
	Arthritis patients with weak HGS	16.086 (13.524–19.134)	4.608 (3.768–5.636)	4.596 (3.758–5.62)
	*p*-value	<0.0001 *	<0.0001 *	<0.0001 *
Self-care	Healthy control with normal HGS	1 (ref.)	1 (ref.)	1 (ref.)
	Healthy control with weak HGS	5.233 (3.995–6.855)	2.49 (1.808–3.429)	2.466 (1.788–3.402)
	Arthritis patients with normal HGS	5.481 (3.981–7.547)	2.619 (1.836–3.734)	2.652 (1.857–3.787)
	Arthritis patients with weak HGS	12.349 (9.411–16.205)	3.872 (2.754–5.445)	3.866 (2.748–5.438)
	*p*-value	<0.0001 *	<0.0001 *	<0.0001 *
Usual activity	Healthy control with normal HGS	1 (ref.)	1 (ref.)	1 (ref.)
	Healthy control with weak HGS	4.106 (3.373–4.998)	2.043 (1.626–2.567)	2.036 (1.62–2.56)
	Arthritis patients with normal HGS	6.343 (5.188–7.754)	3.066 (2.451–3.836)	3.088 (2.466–3.868)
	Arthritis patients with weak HGS	11.744 (9.436–14.615)	3.775 (2.942–4.844)	3.763 (2.934–4.826)
	*p*-value	<0.0001 *	<0.0001 *	<0.0001 *
Pain /discomfort	Healthy control with normal HGS	1 (ref.)	1 (ref.)	1 (ref.)
	Healthy control with weak HGS	1.868 (1.651–2.115)	1.426 (1.245–1.633)	1.417 (1.237–1.624)
	Arthritis patients with normal HGS	4.434 (3.864–5.087)	2.898 (2.505–3.351)	2.901 (2.508–3.356)
	Arthritis patients with weak HGS	6.562 (5.514–7.809)	3.663 (2.995–4.481)	3.662 (2.993–4.48)
	*p*-value	<0.0001 *	<0.0001 *	<.00001 *
Anxiety /depression	Healthy control with normal HGS	1 (ref.)	1 (ref.)	1 (ref.)
	Healthy control with weak HGS	1.947 (1.656–2.289)	1.464 (1.229–1.743)	1.46 (1.226–1.739)
	Arthritis patients with normal HGS	3.088 (2.565–3.718)	2.025 (1.656–2.475)	2.029 (1.659–2.481)
	Arthritis patients with weak HGS	3.933 (3.186–4.856)	2.205 (1.734–2.805)	2.196 (1.726–2.794)
	*p*-value	<0.0001 *	<0.0001 *	<0.0001 *

Abbreviations: EQ-5D European Quality of Life Scale-Five, OR odds ratio, CI confidence interval, OA osteoarthritis, HGS handgrip strength. Values are presented as OR (95% CI). Analyzed by multiple logistic regression analysis. * *p* < 0.05. ^†^ Not adjusted. ^‡^ Adjusted for age, sex, BMI, low income, education ≥ 10 years, high-risk drinking, current smoking, and aerobic physical activity. ^§^ Adjusted for age, sex, BMI, low income, education ≥ 10 years, high-risk drinking, current smoking, aerobic physical activity, DM, HTN, and hypercholesterolemia levels.

## Data Availability

The data presented in this study are available on request from the corresponding author. The data are not publicly available due to the policy of Korea.

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
