# Peer review of "Relation between Handgrip Strength and Quality of Life in Patients with Arthritis in Korea: The Korea National Health and Nutrition Examination Survey, 2015–2018"

_medicina, 2022, doi:10.3390/medicina58020172_

Round 1
Reviewer 1 Report
This is a study comparing handgrip strength with EQ-5D in three large groups of people, namely healthy persons (n=11,610), osteoarthritis patients (n=2,155) and rheumatoid arthritis patients (n=316). The authors came to the unsurprising conclusion that weaker handgrip strength is associated with worse EQ-5D scores. The strength of this paper is the large patient number.
Comments
- The ED-5D is not strictly a quality-of-life measure, in spite of its name. It is a patient-reported outcome measure (PROM) and an ‘instrument to describe and value health’ (according to the EQ-5D web site).
- The other work that studied the relationship between handgrip strength and quality of life used SF-12 (Geryk LL, Carpenter DM, Blalock SJ, DeVellis RF, Jordan JM. Clin Exp Rheumatol. 2015 May-Jun;33(3):366-74) and SF-36 (Slatkowsky-Christensen B, Mowinckel P, Loge JH, Kvien TK. Arthritis Rheum. 2007 Dec 15;57(8):1404-9).
- What is the advantage of having three models? Why not only have one model that includes all the independent variables?
- In arthritis, the handgrip strength may not only reflect muscle weakness or sarcopenia, but can be affected by joint deformity, pain and swelling. The authors should reflect this idea in the discussion.
- This is study of association study rather than cause-and-effect. Poor handgrip strength is associated with poor EQ-5D scores. it does not follow that handgrip strength is due to muscular strength, or that poor EQ-5D index is caused by poor muscular strength. Therefore, it does not follow ‘that strengthening exercise should be considered a tool in improving the overall QOL in patients with arthritis’ (lines 27 and 28).
Author Response
Editor
Medicina
Dear Editor
Subject: Submission of revised paper medicina-1539893
Thank you for your friendly letter enclosing the reviewer’s comments. We have carefully reviewed the comments and have revised the manuscript accordingly. Our responses are given in a point-by-point manner below. Changes to the manuscript are shown in underline/red/bold.
We hope the revised version is now suitable for publication and look forward to hearing from you in due course.
Sincerely.
Seok Kang M.D Ph.D
Department of Physical Medicine and Rehabilitation, Korea University Guro Hospital, Korea University College of Medicine, Seoul, Republic of Korea
Postal address: 148, Gurodong-ro, Guro-gu, Seoul, Republic of Korea
Phone: +82-2-2626-1500
Fax: +82-2-2626-1513
Email: caprock@paran.com
Reviewer #1
- The ED-5D is not strictly a quality-of-life measure, in spite of its name. It is a patient-reported outcome measure (PROM) and an ‘instrument to describe and value health’ (according to the EQ-5D web site).
Response: Thank you very much for your review and opinion. According to your advice and comments, I benefited a lot and added the corresponding revisions in the revised manuscript.
EQ-5D is a widely used global health patient-reported outcome measure that can be applied to a wide range of health conditions. This measure is used to quantify health-related QOL. The greatest strength of this tool is that the subjects can easily respond because there are few questions and it is very simple. There have been many studies on the validity of EQ-5D for the evaluation of health-related QOL in various diseases.
Our study is a cross-sectional observational study based on the KNHANES (Korean National Health and Nutrition Examination Survey). In this nationwide cohort surveillance, quality of life has been evaluated through EQ-5D. Many studies based on KNHANES have used EQ-5D as a health-related QOL measure.
Studies on the feasibility of EQ-5D in the evaluation of the health-related QOL in patients with OA or RA have been reported. The EQ-5D is an acceptable and valid instrument for measuring health-related QOL in patients with OA and RA [1-4].
- The other work that studied the relationship between handgrip strength and quality of life used SF-12 (Geryk LL, Carpenter DM, Blalock SJ, DeVellis RF, Jordan JM. Clin Exp Rheumatol. 2015 May-Jun;33(3):366-74) and SF-36 (Slatkowsky-Christensen B, Mowinckel P, Loge JH, Kvien TK. Arthritis Rheum. 2007 Dec 15;57(8):1404-9).
Response: Thanks for the review
We respect your opinion and indicated in the limitation part that there might be inconsistencies in the QOL evaluation depending on the difference in evaluation tools. [Lines 342-345]
- What is the advantage of having three models? Why not only have one model that includes all the independent variables?
Response: Thanks for the good point.
The advantages of having three models are: There are model 1 unadjusted, model 2 with some variables adjusted (age, sex, BMI, low income, education ≥ 10 years, high-risk drinking, current smoking, and aerobic physical activity), and model 3 with more variable adjusted (in addition to the modified characteristics in model 2 DM, HTN, and hypercholesterolemia levels).
By comparing model 1 and model 2-3, it is possible to find out whether the adjustment of variables itself is meaningful. And in the comparison of model 2 and model 3, it can be confirmed whether the results are affected through more adjusted variables (DM, HTN, and hypercholesterolemia levels).
It has been added to the Statistical analysis in materials and method by reflecting the reviewer's opinion. Lines [143-146]
- In arthritis, the handgrip strength may not only reflect muscle weakness or sarcopenia, but can be affected by joint deformity, pain and swelling. The authors should reflect this idea in the discussion.
Response: Thanks for the good point.
We thoroughly agree with your opinion. It has been added to the discussion based on your comments. Lines [292-294, 328-330]
- This is study of association study rather than cause-and-effect. Poor handgrip strength is associated with poor EQ-5D scores. it does not follow that handgrip strength is due to muscular strength, or that poor EQ-5D index is caused by poor muscular strength. Therefore, it does not follow ‘that strengthening exercise should be considered a tool in improving the overall QOL in patients with arthritis’ (lines 27 and 28).
Response: Thanks for the good point.
We agreed with you that this is study of association study rather than cause-and-effect. So it has been added to the abstract and conclusion based on your comments. Lines [26-27, 353-355]
Reference
- Zrubka, Z.; Rencz, F.; Závada, J.; Golicki, D.; Rupel, V.P.; Simon, J.; Brodszky, V.; Baji, P.; Petrova, G.; Rotar, A.; et al. EQ-5D studies in musculoskeletal and connective tissue diseases in eight Central and Eastern European countries: a systematic literature review and meta-analysis. Rheumatol Int 2017, 37, 1957-1977, doi:10.1007/s00296-017-3800-8.
- Lin, F.J.; Longworth, L.; Pickard, A.S. Evaluation of content on EQ-5D as compared to disease-specific utility measures. Qual Life Res 2013, 22, 853-874, doi:10.1007/s11136-012-0207-6.
- Hurst, N.P.; Kind, P.; Ruta, D.; Hunter, M.; Stubbings, A. Measuring health-related quality of life in rheumatoid arthritis: validity, responsiveness and reliability of EuroQol (EQ-5D). Br J Rheumatol 1997, 36, 551-559, doi:10.1093/rheumatology/36.5.551.
- Bilbao, A.; García-Pérez, L.; Arenaza, J.C.; García, I.; Ariza-Cardiel, G.; Trujillo-Martín, E.; Forjaz, M.J.; Martín-Fernández, J. Psychometric properties of the EQ-5D-5L in patients with hip or knee osteoarthritis: reliability, validity and responsiveness. Qual Life Res 2018, 27, 2897-2908, doi:10.1007/s11136-018-1929-x.
Reviewer 2 Report
Thank you for an interesting manuscript.
My major concern is that Handgrip strength values usually have normative data for understanding age and sex differences. Evidence suggests that handgrip is expected to decrease with advanced age, the average reduction of hand strength per decade ranged between 3–6 kg, in this study the mean age of the control group was 54.87±0.16, and the mean age of the arthritis group was 65.06±0.25, indicating that the arthritis group is about 11 years older (p<.0001). I think that the significant difference in age between the two groups is an important factor to be considered. It would be interesting to demonstrate the HGS values among the two groups (arthritis and control) within similar mean age (p>0.05) for example) where no significant difference between the groups would be recorded according to age. I think this would be feasible as the sample size of the control group is relatively large (n=11,601).
Also, in terms of sex differences, the authors need to elaborate more on the results of HGS related to sex.
Minor comments:
Please elaborate more on the sampling methods.
Please add references for the measurement of HGS and HRQOL.
Author Response
Editor
Medicina
Dear Editor
Subject: Submission of revised paper medicina-1539893
Thank you for your friendly letter enclosing the reviewer’s comments. We have carefully reviewed the comments and have revised the manuscript accordingly. Our responses are given in a point-by-point manner below. Changes to the manuscript are shown in underline/red/bold.
We hope the revised version is now suitable for publication and look forward to hearing from you in due course.
Sincerely.
Seok Kang M.D Ph.D
Department of Physical Medicine and Rehabilitation, Korea University Guro Hospital, Korea University College of Medicine, Seoul, Republic of Korea
Postal address: 148, Gurodong-ro, Guro-gu, Seoul, Republic of Korea
Phone: +82-2-2626-1500
Fax: +82-2-2626-1513
Email: caprock@paran.com
Reviewer #2
My major concern is that Handgrip strength values usually have normative data for understanding age and sex differences. Evidence suggests that handgrip is expected to decrease with advanced age, the average reduction of hand strength per decade ranged between 3–6 kg, in this study the mean age of the control group was 54.87±0.16, and the mean age of the arthritis group was 65.06±0.25, indicating that the arthritis group is about 11 years older (p<.0001). I think that the significant difference in age between the two groups is an important factor to be considered. It would be interesting to demonstrate the HGS values among the two groups (arthritis and control) within similar mean age (p>0.05) for example) where no significant difference between the groups would be recorded according to age. I think this would be feasible as the sample size of the control group is relatively large (n=11,601).
Also, in terms of sex differences, the authors need to elaborate more on the results of HGS related to sex.
Response: We appreciate the review. We agree that the difference in HGS between the groups could be influenced by the significant difference in sex and age.
Our study conducted ANCOVA and multiple logistic regression analysis with data based on the nationwide cohort survey, KNHANES. ANCOVA was performed to control variables that could affect the outcome, including sex, age, etc. Although there was a significant difference between the sex ratio and age of the case and control group, the final results were sufficiently adjusted model of the ANCOVA.
We highly respect your opinion. As per your comment, we performed an age-sex matching analysis (see below Table). There was no significant difference in the overall trend from the original results.
|
Characteristics |
Arthritis (n=1,078) |
Control (n=2,156) |
p-value |
|
Age (years) |
58.74±0.3 |
58.68±0.24 |
0.8794
|
|
Sex |
|
|
0.9552 |
|
Male |
42.02(1.62) |
42.13(1.23) |
|
|
Female |
57.98(1.62) |
57.87(1.23) |
|
|
HGS (kg) |
28.48±0.29 |
29.8±0.24 |
0.0004 |
|
Weak HGS |
27.98(1.5) |
21.1(1.01) |
<.0001* |
|
Low income |
24.15(1.57) |
14.89(0.95) |
<.0001* |
|
Education ≥ 10 years |
51.37(1.77) |
63.64(1.34) |
<.0001* |
|
Smoking |
|
|
0.3951 |
|
Non smoker |
60.36(1.65) |
62.55(1.2) |
|
|
Ex-smoker |
26.24(1.54) |
23.67(1.04) |
|
|
Current smoker |
13.4(1.27) |
13.78(0.94) |
|
|
Drinking |
|
|
0.6301 |
|
Non |
30.56(1.61) |
29.44(1.14) |
|
|
Low risk |
63.26(1.68) |
63.4(1.27) |
|
|
High risk |
6.18(0.91) |
7.16(0.67) |
|
|
Aerobic physical activity |
39.79(1.68) |
42.57(1.29) |
0.1795 |
|
Diabetes mellitus |
15.56(1.35) |
14.51(0.84) |
0.4967 |
|
Hypertension |
40.01(1.69) |
36.87(1.24) |
0.1239 |
|
Hypercholesterolemia |
28.34(1.59) |
29.23(1.18) |
0.6682 |
|
Body mass index (kg/m2) |
24.71±0.13 |
23.87±0.08 |
<.0001* |
|
Waist circumference (cm) |
84.77±0.36 |
82.67±0.23 |
<.0001 |
|
Systolic blood pressure (mmHg) |
121.86±0.56 |
121.24±0.41 |
0.3442 |
|
Diastolic blood pressure (mmHg) |
76.52±0.35 |
76.51±0.25 |
0.9693 |
|
Glucose |
102.93±0.89 |
103.5±0.55 |
0.5736 |
|
Cholesterol |
195.82±1.46 |
197.81±1 |
0.2494 |
|
High-density lipoprotein |
50.59±0.49 |
51.32±0.31 |
0.197 |
|
EQ-5D |
|
|
|
|
Mobility |
29.36(1.51) |
10.16(0.73) |
<.0001* |
|
Self-care |
6.35(0.83) |
2.19(0.33) |
<.0001* |
|
Usual activities |
14.33(1.16) |
4.64(0.52) |
<.0001* |
|
Pain/discomfort |
46.03(1.85) |
20.18(1.01) |
<.0001* |
|
Anxiety/depression |
13.96(1.23) |
8.04(0.72) |
<.0001* |
We would like to reflect your opinion fully, but it seems more appropriate to present the initial analysis results in the manuscript for the following reasons.
First, there are too large numbers of losses in both case and control groups.
Second, KNHANES uses a complex sample design, and the matching method is not used commonly because each person has different statistical weight [1,2]. Even if the matching is the same, the result value is different if the analysis is performed using weight. In the attached Table, it is the result of 1:2 age-sex matching, but the sex ratio and average age do not show the same between case-control groups.
Minor comments:
Please elaborate more on the sampling methods.
Response: Thanks for the good point.
It has been added to the Materials and method based on your comments. Lines [76-79]
Please add references for the measurement of HGS and HRQOL.
Response: Thanks for the good point.
It has been added to the Materials and method based on your comments. Line [118] and Line [126]
Reference
- Kim, Y. The Korea National Health and Nutrition Examination Survey (KNHANES): current status and challenges. Epidemiol Health 2014, 36, e2014002, doi:10.4178/epih/e2014002.
- Kweon, S.; Kim, Y.; Jang, M.J.; Kim, Y.; Kim, K.; Choi, S.; Chun, C.; Khang, Y.H.; Oh, K. Data resource profile: the Korea National Health and Nutrition Examination Survey (KNHANES). Int J Epidemiol 2014, 43, 69-77, doi:10.1093/ije/dyt228.